# Prediction of Incomplete Response of Primary Tumour Based on Clinical and Radiomics Features in Inoperable Head and Neck Cancers after Definitive Treatment

**DOI:** 10.3390/jpm12071092

**Published:** 2022-06-30

**Authors:** Joanna Kaźmierska, Michał R. Kaźmierski, Tomasz Bajon, Tomasz Winiecki, Anna Bandurska-Luque, Adam Ryczkowski, Tomasz Piotrowski, Bartosz Bąk, Małgorzata Żmijewska-Tomczak

**Affiliations:** 1Department of Electroradiology, University of Medical Sciences, 10 Fredry St., 61-701 Poznan, Poland; adam.ryczkowski@wco.pl (A.R.); tomasz.piotrowski@wco.pl (T.P.); bartosz.bak@wco.pl (B.B.); 2Department of Radiotherapy II, Greater Poland Cancer Center, 15 Garbary St., 61-866 Poznan, Poland; tomasz.bajon@wco.pl (T.B.); tomasz.winiecki@wco.pl (T.W.); anna.bandurska-luque@wco.pl (A.B.-L.); 3Department of Medical Biophysics, University of Toronto, 101 College St., Toronto, ON M5G 0A3, Canada; michal.kazmierski@mail.utoronto.ca; 4Department of Medical Physics, Greater Poland Cancer Center, 15 Garbary St., 61-866 Poznań, Poland; 5Department of Radiotherapy I, Greater Poland Cancer Center, 15 Garbary St., 61-866 Poznań, Poland; malgorzata.zmijewska-tomczak@wco.pl

**Keywords:** head and neck cancer, radiotherapy, incomplete response, predictive models, radiomics

## Abstract

Radical treatment of patients diagnosed with inoperable and locally advanced head and neck cancers (LAHNC) is still a challenge for clinicians. Prediction of incomplete response (IR) of primary tumour would be of value to the treatment optimization for patients with LAHNC. Aim of this study was to develop and evaluate models based on clinical and radiomics features for prediction of IR in patients diagnosed with LAHNC and treated with definitive chemoradiation or radiotherapy. Clinical and imaging data of 290 patients were included into this retrospective study. Clinical model was built based on tumour and patient related features. Radiomics features were extracted based on imaging data, consisting of contrast- and non-contrast-enhanced pre-treatment CT images, obtained in process of diagnosis and radiotherapy planning. Performance of clinical and combined models were evaluated with area under the ROC curve (AUROC). Classification performance was evaluated using 5-fold cross validation. Model based on selected clinical features including ECOG performance, tumour stage T3/4, primary site: oral cavity and tumour volume were significantly predictive for IR, with AUROC of 0.78. Combining clinical and radiomics features did not improve model’s performance, achieving AUROC 0.77 and 0.68 for non-contrast enhanced and contrast-enhanced images respectively. The model based on clinical features showed good performance in IR prediction. Combined model performance suggests that real-world imaging data might not yet be ready for use in predictive models.

## 1. Introduction

Treatment of patients diagnosed with inoperable and locally advanced head and neck cancers (LAHNC) remains a challenge for clinicians. Chemoradiation is standard of treatment of these cancers, however, in 5–15% of patients’ incomplete response (IR) of primary tumour is observed [1]. Presence of residual disease negatively affects overall survival and suggests higher aggressiveness of the tumour [2]. Identification of patients with high risk of IR before treatment would be of value to optimization of individual treatment and shared decision-making process.

Apart from clinical factors, imaging-derived features, radiomics, are widely used for prediction of different treatment endpoints including local and regional failure, overall survival (OS) or distant metastases [3,4]. However, performance of models predicting locoregional failure, including very early failure, remains unsatisfactory. Vallieres et al. designed a radiomics-based model predicting locoregional failure in head and neck cancer, based on pre-treatment FDG PET and CT images, which achieved AUROC 0.69 in independent validation for two cohort of patients [4]. Several studies have investigated prediction of overall survival, based on both: clinical and radiomics features, founding that combination of models more accurately predicts local and regional failure as well as OS. Yu et al. designed machine learning model for two-year OS for head and neck cancer patients, which achieved AUROC 0.792 [5]. Similar results were published by Keek and Han [6,7]. Although prediction of survival is one of the most investigated topics in cancer research, prediction of direct result of the treatment, for example persistent disease, would be very important for planning and discussing treatment strategy for individual patient. Apart from clinical factors, imaging features of primary tumour and lymph nodes (LN) are highly investigated [1,8,9]. Model combining clinical and radiomics features for predicting treatment failure in cervical LN were successfully developed and validated by Zhai et al. Their model showed good discrimination of high and low risk group of nodal failure 2 years after treatment, with concordance index (C) of 0.80 [8]. Similar research on prediction of persistent primary disease is scarce.

The amount of data is one of the critical factors in modeling the risk of failure, especially when the number of events is low. High quality data produced in controlled clinical trials might not be sufficient for exploring predictive and prognostic factors due to the relatively low number of highly selected patients included. Moreover, such models are rarely successfully validated on sets of real data, which puts their utility in routine clinical practice into question. Routinely collected data is variable and strongly dependent on the collection procedures, variety of imaging protocols and device vendors; however, real-world databases have an advantage of high number of cases and events, which could potentially compensate for the heterogeneity of collected data. Models based on such databases would be more applicable in daily clinical practice. In this work, we investigated the utility of predictive models for IR based on daily collected clinical data and CT-derived radiomic features. We compared predictive performance of features computed from non-contrast enhanced planning images taken at our center, and contrast enhanced diagnostic scans obtained outside of our center, in local diagnostic centers to investigate if quality of real-world data (RWD) is sufficient for extraction of predictive radiomics features. Leveraging the diverse set of diagnostic scans would enable rapid expansion of the database and accelerate further research.

The aim of the study was to build and evaluate models based on clinical and radiomics features for prediction of incomplete response (IR) after definitive radiotherapy or chemoradiation in patients diagnosed with inoperable and locally advanced squamous cell carcinoma (SCC) of head and neck based on clinical data and routinely collected diagnostic and radiotherapy planning CT images.

## 2. Materials and Methods

The study is a retrospective assessment of outcomes of patients diagnosed with locally advanced SCC of head and neck, treated by definitive radiotherapy or chemoradiation in Greater Poland Cancer Center between January 2011 and December 2021.TRIPOD statement is available in Appendix A. Ethical approval for this study was waived by Ethic Committee of Poznan University of Medical Sciences (KB 367/22) due to retrospective nature of the study (S3).

### 2.1. Patients

290 patients with biopsy-proven SCC of oral cavity, nasopharynx, oropharynx, larynx and hypopharynx were included in the study (Table 1). Patients treated with induction chemotherapy were eligible for the study. We excluded patients with tumours of parotids, thyroid gland and histology other than SCC, and those with prior surgery other than biopsy before radiotherapy. Additionally, cases where the planning CT scan or GTV contours could not be retrieved were also removed from the analysis. All patients underwent diagnostic imaging, including contrast-enhanced CT as well as clinical examination. p16 status was available for 47 patients. In total, 290 patients fulfilled all criteria for the study.

### 2.2. Treatment and Follow Up

Before radiotherapy, both diagnostic contrast-enhanced and non-enhanced CT images of head and neck region were obtained for all patients. For radiotherapy planning purpose slice thickness 3 mm was used.

66 patients were treated with radiotherapy and 224 patients were treated with chemoradiation. Chemoradiation consisted of up to three courses of cisplatin 100 mg/m^2^ on days 1, 22 and 43 or 40 mg/m^2^ weekly up to 6 courses concomitantly with intensity modulated radiotherapy (IMRT). Total planned dose for the tumour was 70 Gy/35 fractions, 60 Gy for high risk volumes and 50 Gy for elective volumes. Primary and nodal Gross Tumour Volumes (GTVp, GTVn respectively) were delineated on non-contrast enhanced CT images, rigidly co-registered with contrast-enhanced diagnostic images.

Clinical Target Volumes (CTV) were adapted individually according to Gregoire et al. [10,11] Planning Target Volumes (PTV) were added to the CTVs as 3D uniform margins of 3 mm according to the set-up and internal motion errors for head and neck treatment, calculated for the institution [12]. All volumes were delineated or reviewed by one radiation oncologist (JK).

After completion of the treatment patients were evaluated by a radiation oncologist and head and neck surgeon 2–4 weeks after treatment, then monthly in the first year after treatment, three-monthly in the second year and every 6 months for the next 3 years. Contrast enhanced CT imaging for radiological response evaluation was performed 3 months after treatment completion. RECIST 1.1. criteria for response evaluation were used to define treatment outcome [13].

### 2.3. Endpoints

Endpoints of the study were incidence of incomplete response (IR), stable disease (SD) or progression (PD) in initial localization of treated tumour, three months after treatment in both imaging and clinical assessment. Residual lesions were confirmed by histopathology in case of surgical salvage resection or biopsy for patients further treated by chemotherapy, immunotherapy or enrolled into a clinical trial. Biopsy was not performed if the lesion was not accessible for these procedures due to its localization or if the patient was a candidate for the best supportive care only. Where possible, FDG PET/CT was performed before surgical salvage procedures.

### 2.4. Clinical Features

The included clinical features were: ECOG performance status, age, gender, clinical stage of tumour according to 7th AJCC edition, localization of primary site and T stage, chemotherapy, p16 status, primary tumour volume (GTVp) and dose delivered. p16 status was coded as positive, negative or unknown in cases where it was not determined. Tumour volume was used as a continuous variable in all of the models to avoid having to select arbitrary cut-off values for low/high volume.

### 2.5. Radiomic Feature Extraction

Radiomic features were extracted using Pyradiomics 2.2.0. We considered the following feature classes: shape, first order, Grey Level Cooccurence Matrix (GLCM), Gray Level Dependence Matrix (GLDM), Gray Level Run Length Matrix (GLRLM), Gray Level Size Zone (GLSZM), Neighbouring Gray Tone Difference Matrix. Extraction was performed on the original image, as well as images filtered using wavelet, square, square root, exponential, logarithm, gradient and Laplacian of Gaussian (LoG) filters. The images were pre-processed by resegmentation to [−600, 150] HU range and fixed bin width discretization with bin width of 25 HU. We also performed interpolation to either isotropic 1 mm or 2 mm spacing before extraction and combined features from both scales. In total, we computed 3190 features from each image [14].

### 2.6. Model Training and Validation

All analysis was performed using Python 3.7. To predict the risk of IR, we trained L1-penalized (Lasso) logistic regression using either clinical variables alone or in combination with the CT image features. Univariate p-values for the clinical features were computed using the F-test for classification. To reduce the dimensionality of radiomic features we first performed unsupervised feature selection by removing features with near-zero variance followed by single-linkage feature agglomeration based on Pearson correlation with threshold of 0.9. The final set of features was selected by maximizing the mutual information between features and targets (presence/absence of IR). The number of selected features and the L1 penalty coefficient for logistic regression were tuned using grid search with cross validation. Model performance was evaluated using the area under the ROC curve (ROC AUC) computed from 5-fold cross validation. Training and evaluation were performed using scikit-learn 0.22.1 [15].

The software source code is available in Appendix A.

## 3. Results

### 3.1. Patients’ Characteristics

Initially 330 patients were included in the study, 290 patients were eligible. Reasons for exclusion were: radiotherapy planning based on megavoltage computed tomography (MVCT) due to dental filling artefacts, too short follow-up due to patient’s death or lost to follow-up for unknown reason. 55 patients (19%) did not receive planned dose of cisplatin due to worsening of performance status or treatment toxicity. All patients included in the study completed radiotherapy course as planned.

Median follow up for whole group was 33.2 months, (range 3–112 months). 45 (15.6%) patients completed treatment without complete remission including 26 (9%) with residual primary tumour, 11 (3.8%) with residual metastatic LN and 8 (2.8%) patients with both. (Table 2) We did not observe any progression or stable disease. 18 primary lesions and 8 residual LN were biopsy or histopathology proven. All lesions considered as IR were localized in irradiated high dose volumes. In 19 (13 primary and 6 LN) cases biopsy was not performed due to location of residual tumour inaccessible for biopsy or surgery, poor performance status and patient’s eligibility to best supportive care only or loss to follow up. In 3 cases IR was confirmed by FDG PET/CT. The 2-year OS rate was 71%.

### 3.2. Clinical Model

Most important clinical features in univariate analysis were primary site oral cavity (*p* < 10^−5^), tumour volume (*p* < 0.001), performance status ECOG higher than 0 and tumour stage T3/4 (both *p* < 0.01) (Figure 1 and Figure 2).

Both models: clinical and combined with radiomics features extracted from non-contrast enhanced images showed good performance with AUCROC 0.78 and 0.77 respectively. Combined model did not improve performance of model based on clinical features only. Radiomics features derived from contrast-enhanced images combined with clinical model decreased its performance to AUROC 0.68 (Figure 3).

## 4. Discussion

In this study we developed and tested clinical and radiomics models for prediction of incomplete remission of primary tumour in patients diagnosed with inoperable cancer of head and neck region, treated with definitive radiotherapy or chemoradiation. Due to noninvasive nature of image analysis, radiomics became a promising tool not only in the evaluation of the risk of failure in head and neck [1,4,6] but also as a tool for tumour segmentation and analysis of treatment-resistant sub-volumes of the tumour [16,17].

There is growing evidence for good performance of radiomics and combined predictive models in prediction of locoregional failure in cancers of head and neck region. The seminal study of Aerts et al. proved that radiomics signature is correlated with treatment outcome and associated with tumour gene expression patterns [18]. Further studies analysed ability of radiomics signatures to predict overall survival as well as treatment failure. Models for nodal failure prediction were designed and validated by Zhai et al., showing good performance [8,9]. Our study presents attempt to evaluate risk of IR based on clinical and radiomics features and results are in line with findings published by others for risk of local failure [4,6]. The model designed and developed in our study, combining clinical and radiomics features and based on non-contrast enhanced CT images performed almost equally well in comparison with clinical model (AUROC 0.78 vs. 0.77 respectively Possible reasons why combining clinical and radiomics features did not improve model performance are complex. One of the reasons might be that the small size of the dataset and low number of events observed was insufficient to address both heterogeneity of clinical data—for example different primary site localization and T stages—as well as the radiomics framework which might not cover all image patterns. A possible solution to the latter problem could be the implementation of deep learning as proposed by Diamant et al. or Le at al. [3,19].

In the clinical model developed in this study, the most important factors for incomplete remission were tumour dependent: localization of primary tumour in oral cavity, tumour volume and T stage. While contrast enhanced CT plays an important role in defining the T stage, the staging information used in the clinical model was based not only on CT scans but also on clinical examination and other imaging modalities like MR or US. Clinical examination is critical for assessing the mobility of vocal cords or mucosal spread, that are both invisible in CT and can only be detected through the physical examination. Assessment of the abovementioned manifestations might cause upstaging of the tumors of the larynx or the oral cavity. MR imaging can contribute to the overall assessment of the T stage and is used in most cases of nasopharyngeal cancer and for soft tissue evaluation. Furthermore, the p16 status affects the staging and prognosis in oropharyngeal cancer. Therefore, the final T stage in the clinical model represents a comprehensive evaluation of the tumour. In contrast, radiomics features are solely image-derived and related, among others, to volume and shape of the tumour, but not directly to the T stage. Moreover, hetrogeneity related to non-harmonized contrast enhanced CT images mentioned above added unexpected noise to the radiomics features.

The 39.3% rate of IR was the highest among patients diagnosed with oral cavity cancer what confirms findings that inoperable oral cavity tumours are relatively resistant to non-surgical treatment [20,21,22]. Performance status consist a well-known factor affecting overall survival prognosis. In our study we confirmed that performance score greater than ECOG 0 is an important factor for IR too. Although majority of patients treated in this study presented ECOG 1 and were initially eligible for chemoradiation, the outcome might be affected by frailty not detected before treatment, including suboptimal nutrition, weight loss, comorbidities [23] as well as heavy smoking and alcohol use. In our department frailty evaluation is performed before treatment as a standard for patients 70 years old and older. Deterioration of performance status during treatment often results in withdrawal of concomitant chemotherapy that results in suboptimal dose of cisplatin during radiotherapy. In our cohort 55 patients (19%) did not received planned dose of cisplatin due to worsening of performance status or treatment toxicity. However, neither chemotherapy nor cumulative dose of cisplatin was a significant clinical factor for incomplete. remission in our cohort. All patients completed radiotherapy course as planned. Patients who did not complete radiotherapy were excluded from the study to avoid introducing the additional confounding factor.

Role of HPV infection in oropharyngeal cancer is well known as a favourable factor for treatment outcome in oropharyngeal cancer [24]. p16 testing is nowadays standard of care, however, in our retrospective group, only 47 patients were p16 tested as p16 testing was not a part of the standard care. Since 2017 p16 immunostaining in oropharyngeal cancer is a part of standard procedure in our center. 29 of tested patients were p16 positive, including 23 patients with oropharyngeal cancer.

Presented clinical model, based on patients’ and disease’s features, can be helpful in estimating of the risk of very early treatment failure and in process of informed shared decision making.

In the spirit of leveraging real-world data, similarly to the recent study in breast cancer [25], we attempted to use not only the standard, non-contrast enhanced CT images acquired in the treatment planning process, but also a heterogenous set of contrast-enhanced CT diagnostic images obtained in our center and other hospitals shortly before treatment.

Thanks to significantly higher availability, using these diagnostic images could help to build larger data sets, provided that the balance between quality and quantity is maintained. It would also help to improve the robustness and generalization of radiomics models, supporting their wider adoption and use [25,26]. Unfortunately, real-world contrast enhanced CT images collected in this study were too heterogeneous to build a predictive radiomics model with satisfactory performance, enabling further clinical testing and use. The heterogeneity of these CT images is caused by the variation in the image acquisition protocols between centers, including varying slice thickness, reconstruction kernel, as well as different kV and mAs. Moreover, we observed variations in volumes of iodine-contrast administered in different departments. These cross-center differences significantly affected the model performance. Solutions for harmonization of CT images data, for example proposed by Selim et al. [27] would provide valuable help in building large scale images database.

The study has some limitations. Not every IR was available for biopsy or pathology examination. For example, residual disease localized in the retropharyngeal space or invading large vessels is typically not safely accessible, especially shortly after chemoradiation. In such cases, diagnosis of IR was made based on radiological assessment of primary tumour localization, contrast enhancement, clinical evaluation of the patient including examination under anaesthesia if needed, and presence of symptoms. Moreover, discrimination between posttreatment changes such as oedema or fibrosis and persistent tumour is often challenging without pathological examination. Due to difficulties in labelling these images as IR or posttreatment changes we didn’t include these patients to the study, unless FDG PET/CT confirmed persistent disease.

## 5. Conclusions

The predictive model based on clinical data collected in routine head and neck clinical practice reached good accuracy. In the future, we plan to prospectively test our approach on a new patient cohort in our clinic, as well as to perform external validation on data from multiple institutions. We continue to work on further use of real-world imaging data for head and neck cancer, as progress in standardization of imaging devices enables obtaining more homogenous and higher quality imaging data.

## Figures and Tables

**Figure 1 jpm-12-01092-f001:**
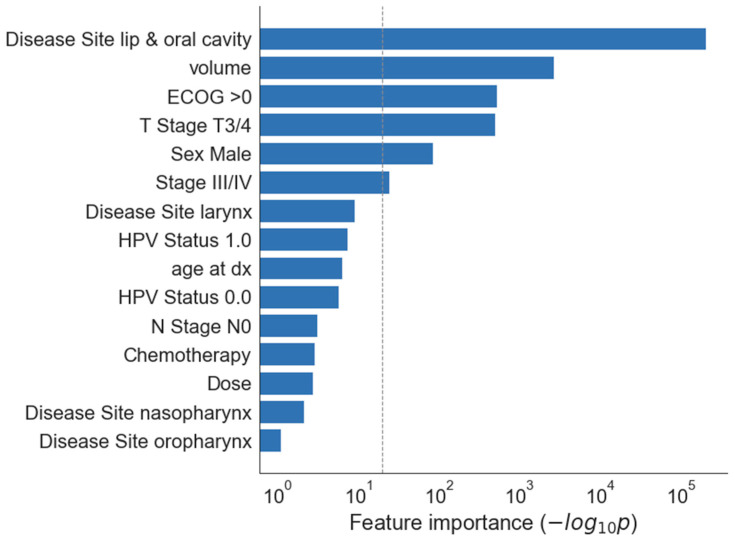
Univariate feature importance for clinical variables. The horizontal axis shows feature importance as the negative log of univariate p value (F-test), so that longer bar indicates a feature more significantly associated with the outcome. The dashed grey line indicates *p* = 0.05. Volume-GTV primary volume, Chemotherapy—yes, Dose—radiotherapy dose delivered, HPV status-1.0—positive, 0.0—negative.

**Figure 2 jpm-12-01092-f002:**
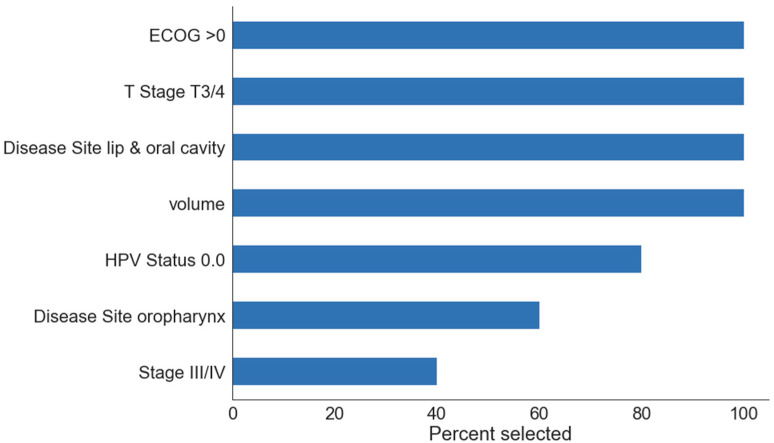
Feature importance in the multivariate clinical model. The importance score is computed as the percentage of cross-validation folds where the feature was selected in the model, so that a score of 100% indicates that the feature was selected in every fold. For clarity, only features that were selected at least once are shown. Volume—GTVp volume, HPV status 0.0—negative.

**Figure 3 jpm-12-01092-f003:**
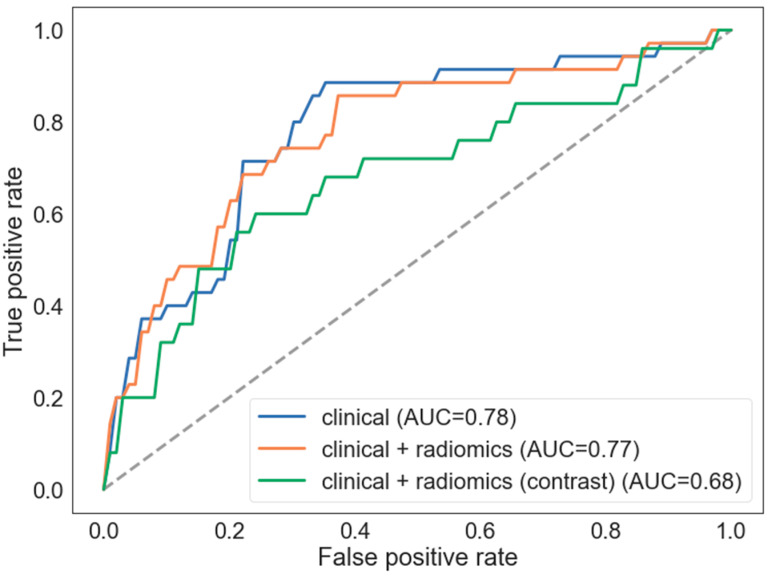
Performance of the clinical only and combined clinical-radiomics model based on non-contrast enhanced images. The AUROC values are averaged over 5 cross-validation folds. Adding radiomics to clinical features did not improve performance (AUROC 0.78 vs. 0.77 respectively).

**Table 1 jpm-12-01092-t001:** Patients’ characteristics.

Characteristic *n* = 290	Value
Age (years)	
Range	20–81
Median	58
Gender	
Male	217
Female	73
Primary Site	
Nasopharynx	17
Oropharynx	131
Hypopharynx	32
Oral cavity	28
Larynx	82
Tumour classification	
T1	15
T2	93
T3	92
T4	90
Tumour Volume (cc)	
Median	13.8
Range	(0.2–91.3)
Stage AJCC v.7	
I	11
II	33
III	66
IVA	173
IVB	7
Follow up (months)	
Median FU	33.2
Range	3–112
HPV status:	
Positive	29
Negative	18
Unknown	243
ECOG 0	79
ECOG 1	211

**Table 2 jpm-12-01092-t002:** Treatment and outcome.

Treatment and Results, *n* = 290	Number of Patients (%)
Treatment	
RT	66 (22.7)
RTCT	224 (77.2)
Residual disease	
All	45 (15.6)
Primary site	26 (9)
Lymph nodes	11 (3.8)
Both	8 (2.8)
Primary site	Primary site residual disease, *n* = 34 (% of all patients, % of all patients in corresponding primary site)


Oropharynx	15 (5.2, 11.4)
Oral cavity	11 (3.8, 39.3)
Larynx	6 (2.1, 7.3)
Hypopharynx	1 (0.3, 3.1)
Nasopharynx	1 (0.3, 5.9)

## Data Availability

The data presented in this study are available on reasonable request from the corresponding author (J.K.).

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
