# Peer review of "Prediction of Incomplete Response of Primary Tumour Based on Clinical and Radiomics Features in Inoperable Head and Neck Cancers after Definitive Treatment"

_jpm, 2022, doi:10.3390/jpm12071092_

Round 1

Reviewer 1 Report

The authors in their material deal with the prognostic values of clinical and radiological (radiomimic) factors of head and neck cancer patients after definitive radiotherapy (RT). The material is really interesting and finally I would like to suggest it to publish. I have some provincial comments and notifications, I have one really important recommendation to think about it and I suggest to expand a little bit the “Discussion” section.

1, Line2: The authors emphasized that they measured the reaction of the primary tumour, however in the final evaluation they described the reaction of the nodal manifestations as well. I do not think that there is necessity to emphasize the primary “property” of the disease (neither in the title).

2, Line14, 35 etc.: Partial response is the part of (good) Response Rate in clinical trials and generally it is a relatively good news in the clinical practice after the introduction of any medical treatment of advanced cancer diseases. Nevertheless, considering RT of head and neck cancers the only partial response generally forecasts the later progression. To resolve this terminology differences I suggest the use of incomplete response in the material.   

3, Line85: Please sign the name of the Cancer Center that contributed.

4, Line126: The performance states of the patients were ECOG 0-1 (before RT). In case of relapse over salvage surgeries no patient receives palliative chemotherapy/biotherapy/immunotherapy, only palliative care?

5, Line143: 3190 radiology features were measured finally in every patients/images. The final conclusion of the work was that the radiomimic factors did not add relevant value to the prognostic score. My basic question: there was no radiology factor (e.g. inhomogeneity, central necrosis) that influence the final results?   

6, Line 163: The overall results of RT was really excellent, only 15.6% of the patients had incomplete response. However, the 2years OS rate was only 71%. Please explain this high number of early mortality.

7, Line 177: Tumour volume is (not surprisingly) a relevant clinical factor that determine the results of therapy. However, in the Methods part there is no definition of low-(intermediate)-high (primary or nodal) tumour volume.

8, Line 188: Radiotherapy dose naturally is also an important prognostic factor. Nevertheless I did not find any data about the percent of incomplete RT courses.

9, Line 208: I think some other authors found any relevance of radiomimic data in prognosis of cancer disease. Furthermore I feel a little bit too short the discussion of this really important issue.

10, Line 220: The deterioration of the general state of the patients during RT of head and neck cancers unfortunately is not a rare finding. It would have been interesting to analyse this factor considering the incomplete result of the therapy. Maybe it would be a great work in the “Results” section, but please describe some more details of your work considering this issue in the “Discussion” section.

11, Table 1:  There was really no patient with ECOG 2 state (able to self-care, but unable to work)?   

12, Line 118: The CT (and clinical) evaluation of the RT results was carried out 3 months after RT. Some tumours response after 3 months observation. Please sign if there was any patient who react to RT after this period.         

Reviewer 2 Report

Unfortunate result, however I took your study as sincere.

Do you have any more ideas as to why the results of the combined model were not good?

In your study, the heterogeneity of the contrast-enhanced CT was the reason for the poor results. What exactly could this heterogeneity be?

According to your results, T-stage is one of the most important clinical feature. And usually, contrast-enhanced CT is also used to determine T-stage. So how did this discrepancy arise?
